# Mechanochemical Upcycling of Waste Polypropylene into Warm-Mix Modifier for Asphalt Pavement Incorporating Recycled Concrete Aggregates

**DOI:** 10.3390/polym16172494

**Published:** 2024-08-31

**Authors:** Jingxuan Hu, Xueliang Jiang, Yaming Chu, Song Xu, Xiong Xu

**Affiliations:** 1School of Materials Science and Engineering, Wuhan Institute of Technology, Wuhan 430205, China; hujingxuan@witpt.edu.cn; 2School of Chemical and Materials Engineering, College of Post and Telecommunication of WIT, Wuhan 430073, China; 3School of Transportation and Logistics Engineering, Wuhan University of Technology, Wuhan 430063, China; 4College of Civil Engineering, Fuzhou University, Fuzhou 350108, China; xusong@fzu.edu.cn; 5Hubei Provincial Engineering Research Center for Green Civil Engineering Materials and Structures, Wuhan Institute of Technology, Wuhan 430073, China; xxucea@wit.edu.cn; 6School of Civil Engineering and Architecture, Wuhan Institute of Technology, Wuhan 430073, China

**Keywords:** waste concrete, waste polypropylene (PP) plastics, warm-mix asphalt modifiers, recycled concrete aggregate, asphalt mixture, road performance

## Abstract

To solve the problems on resource utilization and environmental pollution of waste concrete and waste polypropylene (PP) plastics, the recycling of them into asphalt pavement is a feasible approach. Considering the high melting temperature of waste PP, this study adopted a thermal-and-mechanochemical method to convert waste PP into high-performance warm-mix asphalt modifiers (PPMs) through the hybrid use of dicumyl peroxide (DCP), maleic anhydride (MAH), and epoxidized soybean oil (ESO) for preparing an asphalt mixture (RCAAM) containing recycled concrete aggregate (RCA). For the prepared RCAAM containing PPMs, the mixing temperature was about 30 °C lower than that of the hot-mix RCAAM containing untreated PP. Further, the high-temperature property, low-temperature crack resistance, moisture-induced damage resistance, and fatigue resistance of the RCAAM were characterized. The results indicated that the maximum flexural strain of the RCAAM increased by 7.8~21.4% after using PPMs, while the sectional fractures of the asphalt binder were reduced after damaging at low temperature. The use of ESO in PPMs can promote the cohesion enhancement of the asphalt binder and also improve the high-temperature deformation resistance and fatigue performance of the RCAAM. Notably, the warm-mix epoxidized PPMA mixture worked better close to the hot-mix untreated PPMA mixture, even after the mixing temperature was reduced by 30 °C.

## 1. Introduction

With the rapid progress of urbanization, construction, and transformation in China, the problems of construction waste pollution and garbage siege urgently need to be solved [1,2]. At present, China’s annual output of construction waste is as high as 1.8 billion tons, but the resource utilization rate is less than 5% [3,4,5]. Construction waste mainly includes waste residue, waste bricks and tiles, waste concrete, and other wastes, of which waste concrete accounts for the highest proportion [6,7,8]. The main treatment method is to transport this waste directly to open-air storage or simple landfills in urban suburbs without classification. It can rarely be effectively reused in construction projects, which will not only seriously pollute the ecological environment, but also cause a large amount of waste non-renewable stone resources [9,10,11]. At the same time, the rapid development of road engineering will also lead to the large consumption and increasing shortage of natural non-renewable mineral resources [12,13,14]. Therefore, it is urgent and necessary to fully consider how to effectively replace natural aggregate (NA) with waste concrete aggregate in a green and sustainable way.

As a renewable resource, existing studies have shown that waste concrete can be converted into recycled concrete aggregate (RCA) after crushing and screening, and applied to an asphalt mixture with good application prospects, which can not only reduce garbage accumulation and solve environmental pollution problems but also significantly improve the high-temperature performance and rutting resistance of asphalt pavement [15,16,17]. For example, Wong et al. [18] studied the feasibility of replacing granite aggregate in hot-mix asphalt with waste concrete aggregate. The results showed that the addition of RCA can significantly improve the elastic modulus, creep resistance, and high-temperature performance of an asphalt mixture. Nwakaire et al. [19] studied the influence of RCA content on the road performance of an asphalt mixture. The results stated that when the RCA content is in the range of 60~70%, the freeze–thaw splitting strength ratio and residual stability of the specimens reach the maximum, and they have good high-temperature rutting resistance.

Compared to natural aggregate (NA), RCA has the characteristics of a high crushing value, water absorption, and porosity. Other than this, a large amount of old cement mortar is adhered to the surface of NA, which causes a weak interface strength between asphalt and RCA when applied to asphalt pavement [20,21,22]. Especially under water immersion, the water damage resistance of an asphalt mixture containing RCA will become very poor, which leads to the limit of the large-scale application of RCA in asphalt pavement [23,24,25]. For example, Zhang et al. [26] studied the effect of the interfacial transition zone between RCA and asphalt mortar on the performance of a hot-mix asphalt mixture and demonstrated that the use of RCA can improve the high-temperature performance of an asphalt mixture, but it can reduce its low-temperature crack resistance and water stability. Therefore, the improvement in the water stability of asphalt mixtures containing RCA is the key issue to promote the high-quality application of RCA at a large scale.

A large number of laboratory and field practices proved that the recycling of waste polypropylene (PP) plastic as a modifier into asphalt pavement is a feasible method, which can not only improve the high-temperature performance, rutting deformation resistance, and water stability of asphalt mixtures but can also significantly improve the adhesion of a binder to aggregates. The potential to use PP to solve the insufficient water damage resistance of RCA in asphalt pavement has been proven [27,28,29,30]. For example, Mashaan et al. [31] evaluated the high and low-temperature performance, water damage resistance, and fatigue performance of a waste PP-modified asphalt mixture and found that the waste PP modifier can significantly improve the high-temperature rutting resistance, water stability, and fatigue properties of asphalt mixtures.

However, due to the extremely high melting temperature of waste PP, it is not easy to achieve uniform dispersion in asphalt at the traditional blending temperature (150–160 °C), which will also cause environmental problems such as asphalt smoke emission [32,33,34]. Therefore, it is of great significance to look for effective treatment methods to reduce the melting temperature of waste PP and improve the property uniformity of its modified asphalt binder to address the moisture damage issue of asphalt mixtures containing RCA.

To achieve this goal, this study considers using a thermal-and-mechanochemical method to convert waste PP into high-performance warm-mix asphalt modifiers (PPMs) through a hybrid use of dicumyl peroxide (DCP), maleic anhydride (MAH), and epoxidized soybean oil (ESO) for preparing an asphalt mixture containing RCA (RCAAM). For the prepared RCAAM containing PPMs, the mixing temperature will be decreased by 30 °C as compared to the hot-mix temperature of the RCAAM containing untreated PP. Further, the high-temperature property, low-temperature crack resistance, moisture-induced damage resistance, and fatigue resistance of the RCAAM will be characterized. Figure 1 displays the flowchart of this research plan.

## 2. Materials and Methodology

### 2.1. Raw Materials

#### 2.1.1. Asphalt Binder

The asphalt binder used in this study was virgin bitumen with a Pen. 70 grade which was supplied from a local factory in Wuhan city, China. Its main physical properties were measured, as presented in Table 1.

#### 2.1.2. Coarse and Fine Aggregates

The coarse recycled concrete aggregate (RCA) used in this study was obtained from a waste concrete block produced by the demolition of local houses in Wuhan after crushing and screening. The fine natural aggregate (NA) was limestone provided by Zhongjian Commodity Concrete Co., Ltd., Hubei, Wuhan, China. Their surface appearance and physical properties results are shown in Figure 2 and Table 2, respectively. 

### 2.2. Preparations of PP Modifiers and Modified Asphalt Binders

The PP modifiers used were derived from waste PP by the hybrid use of dicumyl peroxide (DCP), maleic anhydride (MAH), and epoxidized soybean oil (ESO) following the thermal mechanochemical method as published in our paper [35]. The physical appearances of these selected chemicals are shown in Figure 3. The previous study suggested the optimal mixing percentages of DCP, MAH, and ESO, which were 0.3%, 1.5%, and 5%, respectively, by weight of the waste PP for achieving the best performance of modified binders, as shown in Table 3. Therefore, the best modifiers obtained at these fixed percentages were further adopted to prepare the modified asphalt mixtures for performance validations. In addition, the main technical properties and estimated costs of different PPMs are displayed in Table 4. From simple cost estimates, it is clear that the preparation of PPMs is not highly cost-consuming from the aspect of materials, which can in fact be regarded as an upcycling method.

The preparation of PP modifiers and modified asphalt binders includes the following procedures: firstly, the ready-to-use materials were weighed and premixed according to the mixing design shown in Table 3; the premixes were added into the mixing chamber of a Haake torque rheometer at a mixing temperature of 180 °C with a shear speed of 40 rpm for 5 min; the obtained reaction products were crushed and screened after cooling to be used as modifiers; 4% of PPMs by weight were added to the molten virgin asphalt for swelling for 30 min, for which the temperatures were 180 °C for PPM0, 170 °C for PPM3, and 150 °C for PPM3-g-MAH and PPM3-g-MAH/5ESO; and the blends were further processed at a high shearing speed of 4000 rpm for 30 min to prepare the PPM-modified asphalt binders (PPMAs). Correspondingly, they are denoted as PPMA0, PPMA3, PPMA3-g-MAH, and PPMA3-g-MAH/5ESO, respectively. To be clearer, the preparation flowchart is drawn in Figure 4.

### 2.3. PPMA Mixtures Containing RCA

This study adopted AC-13 as the target aggregate gradation for the preparation of PPMA mixtures, as shown in Figure 5. According to the Marshall mix design method, the optimal asphalt contents for preparing each asphalt mixture was finally determined as 4.9%. It is worth pointing out that the mixing temperatures of PPMA0, PPMA3, PPMA3-g-MAH, and PPMA3-g-MAH/5ESO with coarse RCA and fine NA were 180 °C, 170 °C, 150 °C, and 150 °C, respectively. Correspondingly, these modified asphalt mixtures containing RCA were named RCAAM/PPMA0, RCAAM/PPMA3, RCAAM/PPMA3-g-MAH, and RCAAM/PPMA3-g-MAH/5ESO, respectively.

### 2.4. High-Temperature Wheel Tracking Test

According to AASHTO YP63-05, a wheel tracking test was used to evaluate the rutting deformation resistance of the modified asphalt mixtures at a high temperature. Firstly, rut specimens with dimensions of 300 mm × 300 mm × 50 mm were formed by the rolling method. Secondly, the specimen together with the test mold was placed at room temperature for no less than 48 h. Subsequently, the specimen was insulated in a rut testing machine (Beijing Hanton Science Test Instruments Co., Ltd., Beijing, China) at 60 °C for 5 h, and then the rutting test was carried out for 3600 s with a 42 passes/min rolling speed. Finally, the dynamic stability (DS) of the specimen was calculated according to Equation (1) for evaluating the deformation resistance of the target mixtures.
(1)$$DS=\frac{\left(t_{2}-t_{1}\right)\times N}{d_{2}-d_{1}}\times C_{1}\times C_{2}$$
where *DS* represents the dynamic stability of the asphalt mixture, pass/mm; $d_{1}$ and $d_{2}$ represent to the rut depths at 45 min ($t_{1}$) and 60 min ($t_{2}$), respectively, mm; *N* represents the round-trip wheel speed, at 42 pass/min; and $C_{1}$ and $C_{2}$ represent the respective coefficient of the device type and specimen, normally 1.0.

### 2.5. Low-Temperature Blending Test

According to AASHTO T321, a trabecular bending test was adopted to evaluate the effect of the PPM on the low-temperature cracking resistance of asphalt mixtures. Before the test, prismatic trabeculae with dimensions of 250 mm × 30 mm × 35 mm and a span of 200 mm were placed in a low-temperature environment of −10 °C for 5 h. Then, a universal testing machine (UTM), manufactured from IPC Global Co. Ltd., Sydney, Australia, was used to load the specimens at a single point at a rate of 50 mm/min. Finally, the flexural tensile strength ($R_{B}$), maximum flexural strain ($\varepsilon_{B}$) and flexural stiffness modulus ($S_{B}$) were calculated according to Equations (2)–(4).
(2)$$R_{B}=\frac{3\times L\times P_{B}}{2\times b\times h^{2}}$$
(3)$$\varepsilon_{B}=\frac{6\times h\times d}{L^{2}}$$
(4)$$S_{B}=\frac{R_{B}}{\varepsilon_{B}}$$
where $R_{B}$ represents the flexural tensile strength, MPa; $\varepsilon_{B}$ represents the maximum flexural strain, *με*; $S_{B}$ represents the flexural stiffness modulus, MPa; *b* represents the across width of the interrupt interview piece, mm; *h* represents the across interrupt interview document height, mm; *L* represents the span of the specimen, mm; $P_{B}$ represents the maximum load at failure of the specimen, N; and *d* represents the mid-span deflection of the specimen at failure, mm.

### 2.6. Moisture-Induced Damage Resistance

#### 2.6.1. Immersion Marshall Test

According to ASTM D6927, a Marshall test was used to evaluate the moisture-induced damage resistance of the asphalt mixtures. Firstly, the Marshall specimens were placed in a 60 °C water bath for 30 min and 48 h, respectively. Then, the specimens were tested at a loading rate of 50 mm/min, and their Marshall stability and flow values were recorded. Finally, the residual Marshall stability ($MS_{0}$) of the target samples was calculated according to Equation (5).
(5)$$MS_{0}=\frac{MS_{1}}{MS}\times 100\%$$
where $MS_{0}$ represent the immersed residual stability of the specimen, %; $MS_{1}$ represent the stability of the specimen after moisture immersion for 48 h, kN; and MS represents the stability of the specimen after moisture immersion for 30 min, kN. 

#### 2.6.2. Freeze–Thaw Splitting Test 

According to AASHTO T283, a freeze–thaw splitting test was used to measure the strength ratio of the splitting failure of the specimens before and after water damage to evaluate the water stability of the asphalt mixtures. The test steps are as follows: (1) put the specimens into a vacuum chamber with a vacuum degree of 730 mmHg for 15 min; (2) restore the normal pressure and place the specimens in water for 30 min; (3) freeze the specimens at −18 °C for 16 h; (4) thaw the specimens in a 60 °C water bath for 24 h; and (5) after 2 h in a 25 °C water bath, take the specimens out and perform the splitting test at a loading rate of 50 mm/min, to obtain the maximum load. The splitting tensile strength of the asphalt mixture was calculated according to Equations (6) and (7).
(6)$$R_{T1}=0.006287P_{T1}/h_{1}$$
(7)$$R_{T2}=0.006287P_{T2}/h_{2}$$
where $R_{T1}$ represents the splitting tensile strength of the first group of single specimens without a freeze–thaw cycle, MPa; $R_{T2}$ represents the splitting tensile strength of the second group of single specimens subjected to a freeze–thaw cycle, MPa; $P_{T1}$ represents the test load value of the first group of single specimens, N; $P_{T2}$ represents the test load value of a single specimen in the second group, N; $h_{1}$ represents the height of each specimen in the first group, mm; and $h_{2}$ represents the height of each specimen in the second group, mm.

The ratio of the freeze–thaw splitting strength (*TSR*) of the asphalt mixture was calculated according to Equation (8), and the water stability of the asphalt mixture performed better when the value was tested larger.
(8)$$TSR=\frac{{\overset{¯}{R}}_{T2}}{{\overset{¯}{R}}_{T1}}\times 100\%$$
where *TSR* represents the ratio of the freeze–thaw splitting strength of the specimen, %; ${\overset{¯}{R}}_{T2}$ represents the average splitting tensile strength of the second group of effective specimens after a freeze–thaw cycle, MPa; and ${\overset{¯}{R}}_{T1}$ represents the average splitting tensile strength of the first group of effective specimens without a freeze–thaw cycle, MPa.

### 2.7. Indirect Tensile Test (IDT)

According to ASTM D6931-2017, an IDT was carried out to measure the fatigue life of the compacted asphalt mixtures subjected to repeated bending loads at medium temperatures. For the test, within the limit failure stress (1.3 MPa) of the specimens, two stress values of 0.4 MPa and 0.6 MPa were taken, respectively, and the test load corresponding to the stress value was calculated according to Equation (9) as the loading peak value. A universal testing machine (UTM) was used to load the specimens, the loading mode was semi-sinusoidal cyclic loading with a loading period of 0.5 s, unloading for 0.25 s after each loading of 0.25 s, and the loading frequency and test temperature were 2 Hz and 25 °C, respectively.
(9)$$R_{T}=0.006287P_{T}/h$$
where $R_{T}$ represents the stress value applied to the specimen, MPa; $P_{T}$ represents the test load value of the specimen, N; and *h* represents the height of the specimen, mm.

## 3. Results and Discussion

### 3.1. High-Temperature Performance of RCAAMs Containing PPMs

Figure 6 reflects the effect of PPMs on the rut depth and dynamic stability of the RCAAMs. Correspondingly, the rut depths at 45 min and 60 min are listed in Table 5. As clearly seen, with the use of PPM3 and PPM3-g-MAH instead of PPM0, the final rut depth of the modified asphalt mixture increases from 1.137 mm to 1.512 mm and 2.001 mm, while the dynamic stability decreases from 4228 pass/mm to 3559 pass/mm and 2930 pass/mm, respectively. It is indicated that RCAAM/PPM0 has better resistance to rutting deformation at high temperatures in contrast to RCAAM/PPM3 and RCAAM/PPM3-g-MAH. This is because the mechanochemical degradation and grafting effects can reduce the molecular weight of waste PP, which thus leads to a reduction in the elastic modulus of the modified asphalt binders for deformation resistance. 

Moreover, compared to RCAAM/PPM3-g-MAH/5ESO, the rut depth and dynamic stability of RCAM/PPM3-g-MAH decreases and increases by 0.583 mm and 1298 pass/mm, respectively. It is stated that the epoxidized PPM can better improve the deformation resistance of the RCAAMs compared to grafted PPM. This is because the epoxy groups in the ESO molecular structure can chemically interact with the functional groups that exist in the MAH-based grafted structure to create a newly-formed molecular structure, which contributes to an increase in the elastic modulus, thus significantly improving the high-temperature rutting resistance. It is worth noting that compared to RCAAM/PPM3-g-MAH/5ESO, the rut depth of RCAAM/PPM3-g-MAH/5ESO increases from 1.137 mm to 1.418 mm and the dynamic stability decreases from 4457 pass/mm to 4228 pass/mm, which only decreases by 5.14%, but its mixing temperature decreases by 30 °C. It is proven that warm-mix RCAAM/PPM3-g-MAH/5ESO can still achieve the same equivalent deformation resistance as that of hot-mix RCAAM/PPM0, which causes a reduction in energy consumption and carbon emissions during mixing and paving.

### 3.2. Low-Temperature Cracking Resistance of RCAAMs Containing PPMs

#### 3.2.1. Flexural Strength

Figure 7 presents the effect of PPMs on the *R_B_* value of the RCAAMs. It can be found that after using PPM3 and PPM3-g-MAH for replacement, the *R_B_* value of RCAAM/PPM0 increases from 7.31 MPa to 9.54 MPa and 9.96 MPa, which is an increase of 30.5% and 36.3%, respectively. It is illustrated that RCAAM/PPM3 and RCAAM/PPM3-g-MAH have better damage resistance than RCAAM/PPM0, and their resistance to temperature shrinkage is stronger in low-temperature environments. This is dependent upon the molecular weight of the waste PP which will be reduced by mechanochemical degradation and grafting, and thus the flexibility and deformability of its modified asphalt binder will be improved, leading to enhanced cracking resistance at low temperatures. After a further epoxidation pretreatment of PPM3-g-MAH, the *R_B_* value of RCAAM/PPM3-g-MAH decreases from 9.96 MPa to 7.83 MPa, which falls by 21.4%. It is revealed that ESO can reduce the low-temperature crack resistance of RCAM/PPM3-g-MAH. The reason is that the epoxy groups in ESO will react with the grafted structure of PPM3-g-MAH to produce products with a higher molecular weight, leading to small reductions in the plasticity of the modified asphalt binders and mixtures. 

#### 3.2.2. Maximum Flexural Strain

Figure 8 exhibits the effect of PPMs on the *ε_B_* value of the RCAAMs. It is very clear that the *ε_B_* values of all the modified asphalt mixtures are greater than 2000 με. It is shown that all the RCAAMs have good low-temperature crack resistance. Compared to RCAAM/PPM0, the *ε_B_* values of RCAAM/PPM3, RCAAM/PPM3-g-MAH, and RCAAM/PPM3-g-MAH/5ESO are increased by 15.0%, 21.4%, and 7.8%, respectively. It is manifested that a series of degradation, grafting, and epoxidation pretreatments of waste PP can improve the cracking resistance of the RCAAM at low temperatures. The results obtained can explain that the use of DCP can cause the degradation of waste PP, with massive degradation products of a lower molecular weight, leading to more flexible characteristics in the modified asphalt binder, which can thus improve the low-temperature deformation resistance of the modified asphalt mixture.

### 3.3. Low-Temperature Fracture Macroscopic Morphology of RCAAMs Containing PPMs

Figure 9 displays the low-temperature fracture macroscopic morphology of the RCAAMs. It is clear that a large amount of asphalt separation and a small amount of RCA fracture occur in the RCAAM/PPM0 sectional structure. The fracture structure of RCAAM/PPM3 and RCAAM/PPM3-g-MAH is very flat and is mainly characterized by aggregate fractures, where the asphalt binder is densely filled in the aggregate skeleton. The results state that PPM3 and PPM3-g-MAH can significantly improve the damage morphology of the asphalt–aggregate interface. This is because the original PPMA binder exhibits brittleness at low temperatures and is more prone to brittle fracture before the breaking of the aggregate when subjected to external loads, while degradation and graft modification can improve the low-temperature flexibility of the PPMA, which causes aggregate failures before the addition of the asphalt binder. Compared to RCAAM/PPM3, the fracture structure of RCAAM/PPM3-g-MAH/5ESO is relatively flat, with asphalt separation and aggregate fracture, which indicates that PPM3-g-MAH/5ESO has the benefit of improving the low-temperature flexibility of RCAAM/PPM0, however, not at the level of PPM3 and PPM3-g-MAH.

### 3.4. Moisture-Induced Damage Resistance of RCAAMs Containing PPMs

#### 3.4.1. Residual Marshall Stability

Figure 10 presents the immersion Marshall test results of RCAAMs containing PPMs. It is clear that the Marshall stability of the asphalt mixtures decreases to different degrees after 48 h of immersion in water. It is shown that long-term immersion will reduce the resistance of asphalt mixtures to moisture-induced damage. Compared to RCAAM/PPMA3 and RCAAM/PPMA3-g-MAH, the immersed stability of RCAAM/PPM0 decreases from 12.49 kN to 10.87 kN and 10.62 kN, with decrements of 1.92 kN and 1.87 kN, respectively. When PPMA3-g-MAH/5ESO is used, the initial stability and immersed stability of asphalt mixtures is restored to 13.12 kN and 11.84 kN, respectively. It is demonstrated that after using PPM3 and PPM3-g-MAH, the resistance of RCAAMs to moisture-induced damage decreases, while RCAAM/PPM3-g-MAH/5ESO works almost as well as RCAAM/PPM0.

Figure 11 reflects the effect of PPMs on the Marshall residual stability of RCAAMs. As clearly seen, after using PPMA3 and PPMA3-g-MAH, the MS_0_ value of RCAAM/PPM0 decreases from 91.4% to 86.5%, and 85.6%, which are reductions of 4.9%, and 5.8%, respectively. The results prove that DCP and MAH will reduce the moisture damage resistance of PPMA mixtures. This is because the degradation effect of DCP and the grafting effect of MAH result in a lower viscosity of the PPMA binder, the cohesiveness of the asphalt binder and aggregate is weakened, and the asphalt mixture is more vulnerable to water damage under the conditions of long-term immersion, but their MS_0_ values are greater than 80%, which still meets the technical requirements for the water stability of the mixture. Compared to RCAAM/PPM3-g-MAH/5ESO, the MS_0_ value of RCAAM/PPM3-g-MAH recovered from 85.6% to 90.2%. It is indicated that ESO will significantly improve the water damage resistance of RCAAM/PPM3-g-MAH, and its water stability can be restored to the same level as the hot-mix RCAAM/PPM0. This is because the high-molecular-weight products produced during the epoxidation reaction of PP enhance the adhesion of the asphalt binder to RCA, thus improving the water stability of the modified asphalt mixture.

#### 3.4.2. Freezing–Thawing Splitting Strength Ratio (TSR)

Figure 12 shows the freeze–thaw splitting strength test results of RCAAMs containing PPMs. It is discovered that the freeze–thaw splitting strength of RCAAM/PPM0 is 1.24 MPa, which is 1.09 times and 1.12 times that of RCAAM/PPM3 and RCAAM/PPM3-g-MAH, respectively. After the epoxidation treatment of PPM3-g-MAH by ESO, the splitting strengths of RCAAM/PPM3-g-MAH before and after freeze and thaw recover from 1.32 MPa and 1.11 MPa to 1.39 MPa and 1.20 MPa, respectively, which are close to that of RCAAM/PPM0. It is manifested that the degradation effects of MAH and DCP on waste PP have negative impacts on the freeze–thaw damage resistance of RCAAM/PPM0, but ESO can effectively restore the freeze–thaw damage resistance of RCAAM/PPM3-g-MAH.

Figure 13 presents the effect of PPMs on the freeze–thaw splitting strength ratio of RCAAMs. It can be found that compared to RCAAM/PPM3, RCAAM/PPM3-g-MAH, and RCAAM/PPM3-g-MAH/5ESO, the TSR value of RCAAM/PPM0 decreases from 87.8% to 84.5%, 83.7%, and 86.1%, which are reductions of 3.3%, 4.1%, and 1.7%, respectively. It is stated that a series of degradation, grafting, and epoxidation pretreatments of waste PP will somewhat reduce the water stability of the modified asphalt mixture. It is worth pointing out that the TSR values of all RCAAMs were higher than 75%, which indicates that these mixtures can satisfy the application requirement of reducing damaging risk after freezing and thawing. In addition, it also illustrated that the different thermo-mechanochemical treatments contribute to decreasing the mixing temperature towards warm mix conditions, which will not result in severe freeze–thaw deterioration of the mixtures.

### 3.5. Fatigue Resistance of RCAAMs Containing PPMs

Figure 14 exhibits the effect of PPMs on the fatigue performance of RCAAMs. As clearly seen, the fatigue lives of modified asphalt mixtures under an applied stress of 0.4 MPa were performed significantly longer than those under an applied stress of 0.6 MPa. This result indicates that all the mixtures are very susceptible to an applied fatigue load. Compared to RCAAM/PPM0, the fatigue lives of RCAAM/PPM3 and RCAAM/PPM3-g-MAH present a decreased trend regardless of the applied stress being 0.4 MPa or 0.6 MPa. This result reveals that PPM3 and PPM3-g-MAH have a negative effect on the fatigue resistance of RCAAMs. This is attributed to the fact that mechanochemical degradation and grafting pretreatment will reduce the adhesion of PPMA binders to aggregates, which makes them more prone to micro-cracks between the asphalt and aggregates when subjected to repeated traffic loads. It is worth noting that after further epoxidation pretreatment with ESO, the fatigue life of RCAAM/PPM3-g-MAH at stresses of both 0.4 MPa and 0.6 MPa recovers from 1102 cycles and 175 cycles to 1643 cycles and 366 cycles, respectively. The obtained result indicates that the further epoxidation of ESO to PPM3-g-MAH can to some extent improve the mixture performance to fatigue cracks. This is because the introduction of ESO causes chemical reactions with PPM3-g-MAH to enhance the cohesion of the mixture.

## 4. Conclusions

This study adopted a thermal-and-mechanochemical method to convert waste PP into high-performance warm-mix asphalt modifiers. For the prepared RCAAM containing PPMs, its mixing temperature is about 30 °C lower than that of hot-mix RCAAM containing untreated PP. Further, the high-temperature property, low-temperature crack resistance, moisture-induced damage resistance, and fatigue resistance performance of the RCAAMs were characterized. The main conclusions are as follows:From the high-temperature wheel tracking test results, compared to the untreated hot-mix PPMA mixture, the dynamic stability of the warm-mix epoxidation PPMA mixture decreases by only 5.14% as the mixing temperature and compaction temperature reduce by about 30 °C.The low-temperature bending test results state that after a series of mechanical and chemical pretreatments of waste PP, the maximum bending strain of modified asphalt mixtures increases by 7.8~21.4%, indicating that the low-temperature crack resistance of the mixtures can be improved by the prepared modifiers.The low-temperature fracture morphology demonstrates that the pretreated PPMs can significantly improve the fracture morphology of the asphalt–aggregate interface. The increased low-temperature flexibility of the modified asphalt binder can reduce the risk of brittle fracture in the corresponding mixtures.Water stability test results show that the degradation and grafting of waste PP will have a negative impact on the moisture-induced damage resistance of modified asphalt mixtures, but their MS_0_ and TSR values can still meet application requirements. After the further epoxidation of PPM3-g-MAH, the MS_0_ and TSR values of RCAAM/PPM3-g-MAH can be recovered to a level close to those of RCAAM/PPM0, respectively.Fatigue test results indicate that compared to that of PPMA0, the adhesions of PPMA3 and PPMA3-g-MAH to aggregates are lower and possibly produce more smaller microcracks between the asphalt and aggregates when subjected to repeated traffic loads, but as ESO is added, the fatigue resistance of the modified asphalt mixture can be restored to some degree due to the enhanced cohesions caused by chemical reactions.

## 5. Research Limitations and Future Works

This study provides a technical way to chemically convert waste PP into high-quality warm-mixing asphalt modifiers, which can not only solve the technical problem of the higher mixing temperature of PP-modified asphalt binders and mixtures but also improve their overall engineering performance. Although some valuable research results have been obtained, the research limitations, future research directions, and industrial applications can be considered for formulating further action. They include the following:The main research limitations are the controls of mechanochemical conditions and raw materials for the preparation of PPMs;Future research directions are suggested, focusing on durability validations and rejuvenation explorations; andIndustrial applications can be considered for expressways, first-class or second-class highways, and urban roads, especially where the climate temperature is high.

## Figures and Tables

**Figure 1 polymers-16-02494-f001:**
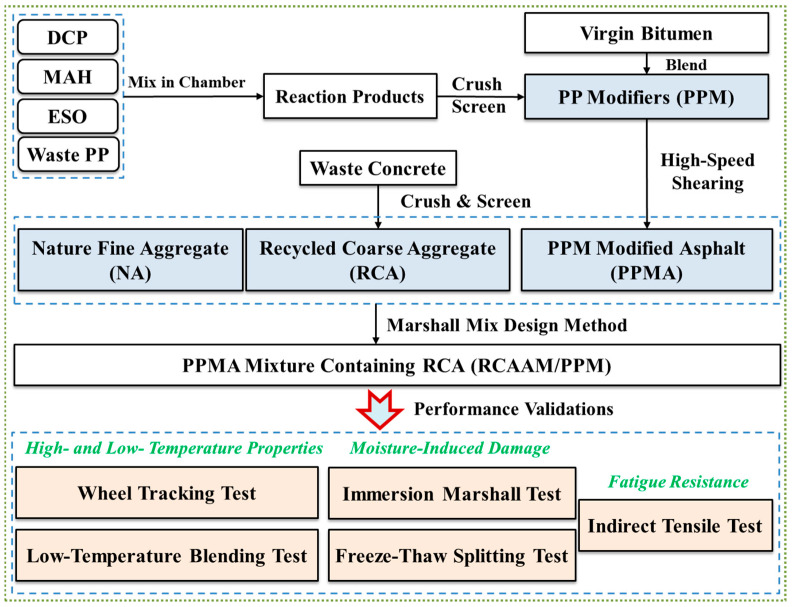
Flowchart of this research.

**Figure 2 polymers-16-02494-f002:**
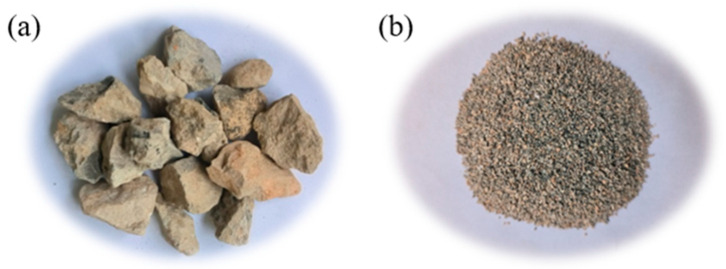
Surface appearance of (**a**) RCA and (**b**) NA.

**Figure 3 polymers-16-02494-f003:**
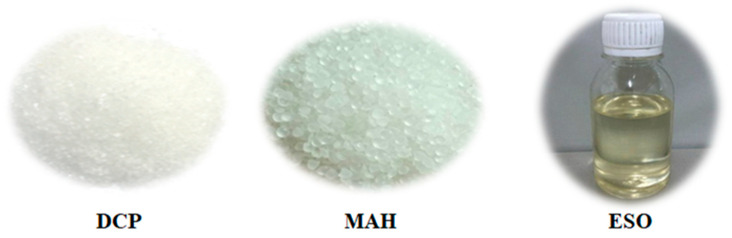
Physical appearance of the used raw materials.

**Figure 4 polymers-16-02494-f004:**
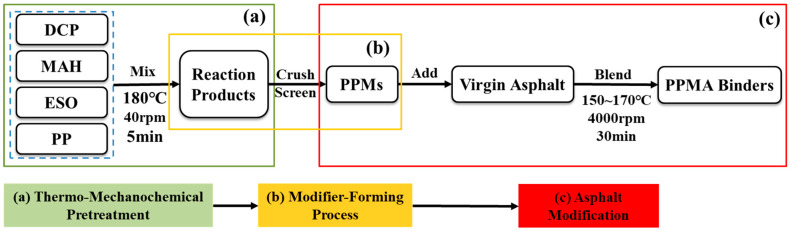
Preparation flowchart of PPMA binders.

**Figure 5 polymers-16-02494-f005:**
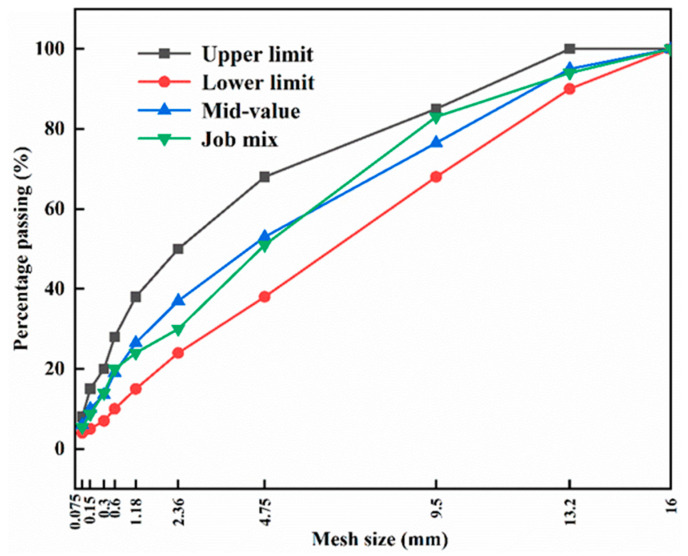
The AC-13 gradation curve for the job mix of the target asphalt mixtures.

**Figure 6 polymers-16-02494-f006:**
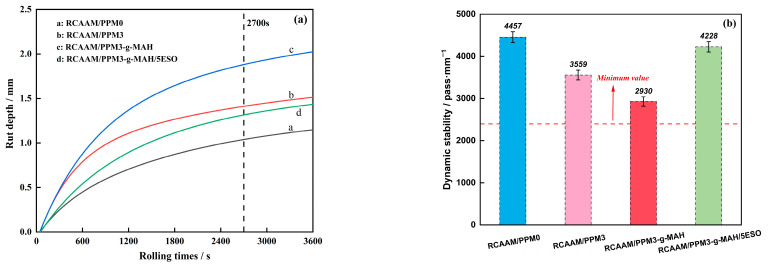
Effects of PPM on (**a**) rut depth and (**b**) dynamic stability of RCAAMs.

**Figure 7 polymers-16-02494-f007:**
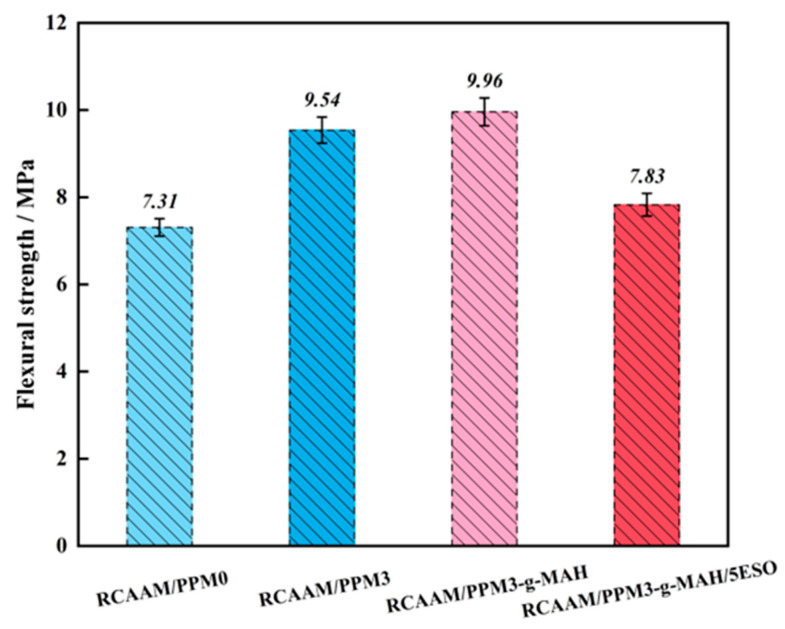
Effects of PPMs on flexural strength (R_B_) of RCAAMs.

**Figure 8 polymers-16-02494-f008:**
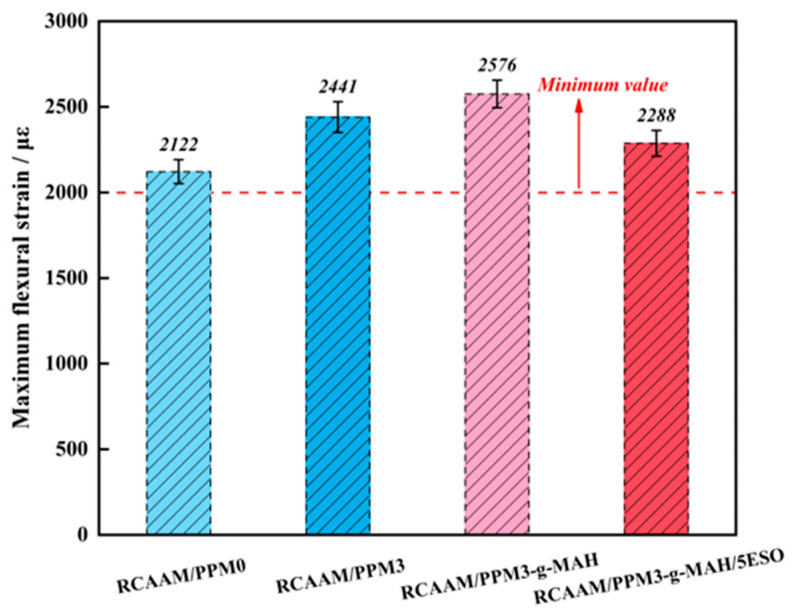
Effects of PPM on maximum flexural strain (*ε_B_*) of RCAAM.

**Figure 9 polymers-16-02494-f009:**
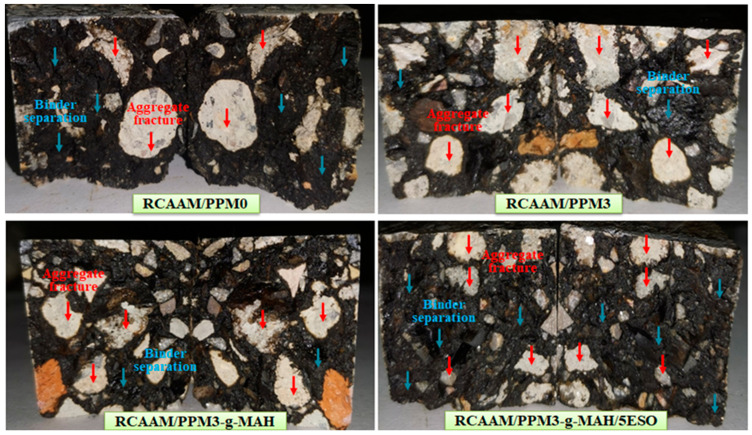
Macroscopic fracture morphology of RCAAMs containing PPMs at low temperature.

**Figure 10 polymers-16-02494-f010:**
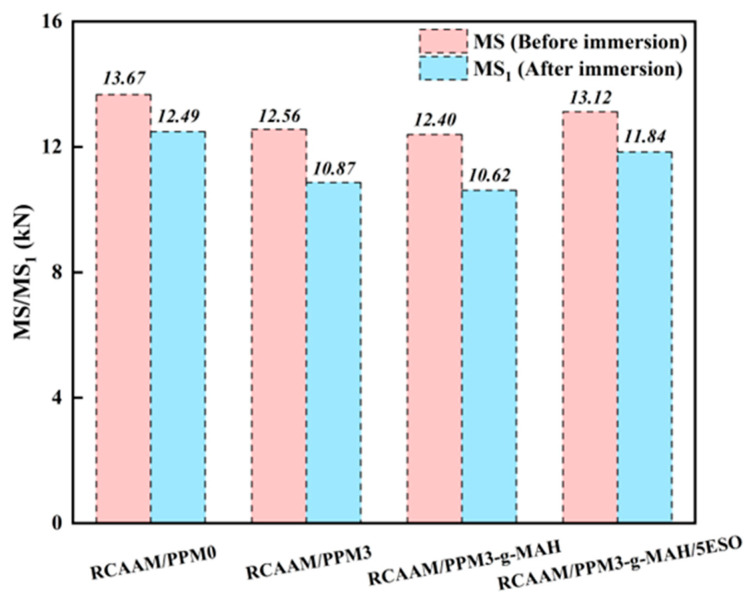
Immersed Marshall test results of RCAAMs containing PPMs.

**Figure 11 polymers-16-02494-f011:**
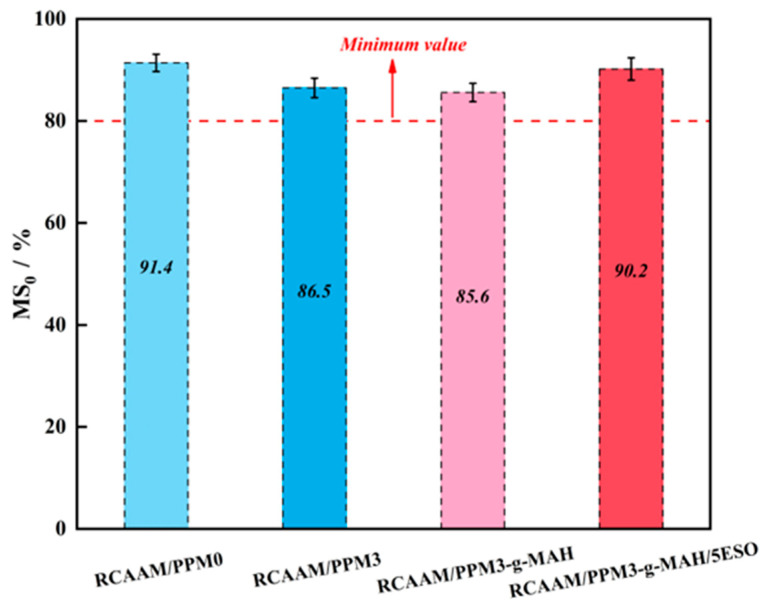
Residual Marshall stability of RCAAMs containing PPMs.

**Figure 12 polymers-16-02494-f012:**
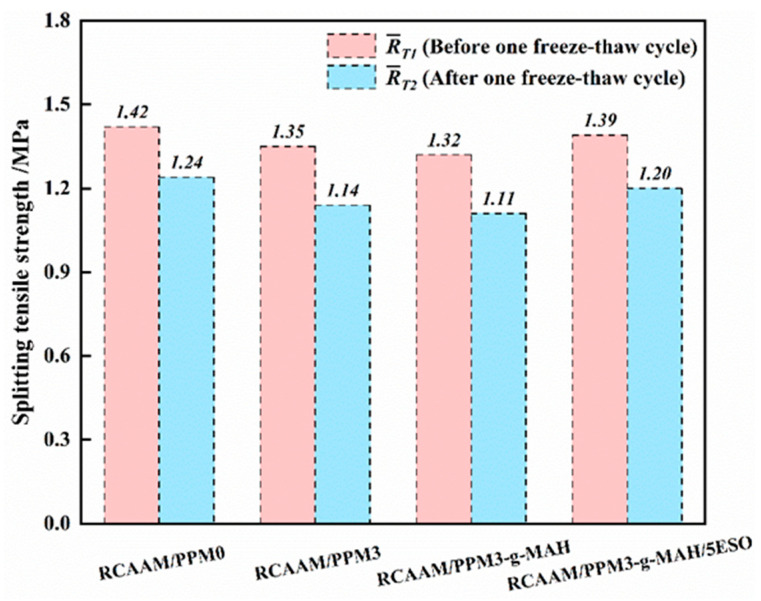
Freeze–thaw splitting test results of RCAAMs containing PPMs.

**Figure 13 polymers-16-02494-f013:**
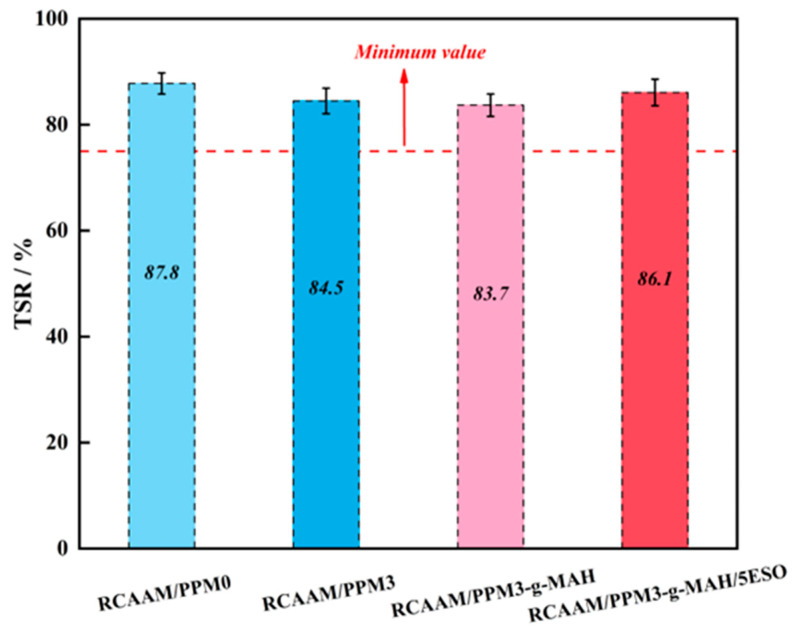
TSRs of RCAAMs containing PPMs.

**Figure 14 polymers-16-02494-f014:**
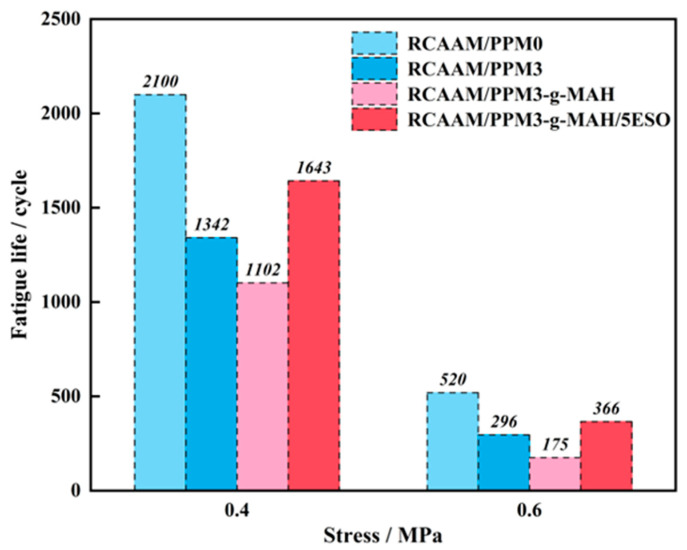
Effect of PPMs on fatigue properties of RCAAMs.

**Table 1 polymers-16-02494-t001:** The main physical properties of the used asphalt binder.

Item	Test Result	Requirement	Standard
Penetration (0.1 mm)	63	60–80	ASTM D5
Softening point (°C)	48.5	≥43	ASTM D36
Viscosity at 135 °C (Pa·s)	0.45	≤3	ASTM D4402
Ductility at 15 °C (cm)	>100	>100	ASTM D113

**Table 2 polymers-16-02494-t002:** Results of physical properties of coarse and fine aggregates.

Aggregate Type	Item	Test Result	Requirement	Standard
Coarse RCA	Apparent density (g/cm^3^)	2.74	≥2.50	ASTM C127
	Los Angeles abrasion loss (%)	17.3	≤28	ASTM C131
	Crushing value (%)	22.6	≤28	ASTM C942
	Water absorption (%)	5.81	≤3.0	ASTM C127
Fine NA	Apparent density (g/cm^3^)	2.71	≥2.50	ASTM C128
	Fine aggregate angularity (%)	42.4	≥30	AASHTO TP33
	Sand equivalent (%)	78.4	≥60	ASTM D2419

**Table 3 polymers-16-02494-t003:** Mix design for preparing PPMs.

	Waste PP	DCP	MAH	ESO
		(By Weight of Waste PP)		
PPM0	100%	0	0	0
PPM3	100%	0.3%	0	0
PPM3-g-MAH	100%	0.3%	1.5%	0
PPM3-g-MAH/5ESO	100%	0.3%	1.5%	5%

**Table 4 polymers-16-02494-t004:** Main technical properties and estimated costs of different PPMs.

Item	Particle Appearance	Meltable Temperature (°C)	Melt Flow Rate (g/10 min)	Estimated Cost * (RMB/ton)
PPM0	White	180	9.94	4000
PPM3	Light yellow	170	125.28	4033
PPM3-g-MAH	Light yellow	150	142.75	4108
PPM3-g-MAH/5ESO	Light yellow	150	109.82	4328

Note: * The estimated cost is calculated only up to the cost of the raw materials regardless of processing, energy, and labor costs. The market prices of waste PP, DCP, MAH, and ESO are approximately 4000 RMB/ton, 11,000 RMB/ton, 5000 RMB/ton, and 4400 RMB/ton, respectively, by August 2024 in China.

**Table 5 polymers-16-02494-t005:** Rut depth of RCAAMs containing PPMs.

Item	t_1_/min	t_2_/min	Δt/min	d_1_/mm	d_2_/mm	Δd/mm
RCAAM/PPM0	45	60	15	1.054	1.137	0.083
RCAAM/PPM3				1.408	1.512	0.104
RCAAM/PPM3-g-MAH				1.894	2.001	0.107
RCAAM/PPM3-g-MAH/5ESO				1.322	1.418	0.096

## Data Availability

Data are contained within the article.

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
