# Peer review of "Mechanochemical Upcycling of Waste Polypropylene into Warm-Mix Modifier for Asphalt Pavement Incorporating Recycled Concrete Aggregates"

_polymers, 2024, doi:10.3390/polym16172494_

Round 1

Reviewer 1 Report

Comments and Suggestions for Authors

In this manuscript, the authors tried to incorporate waste PP into the concrete aggregates, which gave comparable or even better performance. They demonstrated a pathway of PP upcycling. Though the concept is interesting, the authors still need to do further research to strengthen their viewpoint. Thus, I would recommend this work for formal publication after a major revision.

1.      In Figure 1&4, the first arrow is slightly misleading, and the contents are slightly duplicated. For the first arrow, it should indicate the additives were mixed with PP, not “generating” PP.

2.      A horizontal line should be added in Table 2 to distinguish two materials.

3.      In the preparation of PPMs, the authors only studied three recipes, is there any rational in just choosing these three and why the percentages were fixed instead of varied.

4.      DSC measurements should be conducted in the PPMs case to indicate any chemical reactions happening during the melting.

5.      The authors should give a cost estimate for this upcycling approach. At least the peroxide is not super cheap, and they need to state it is a real upcycling process instead of downcycling.

Reviewer 2 Report

Comments and Suggestions for Authors

The manuscript "Mechanochemical Upcycling of Waste Polypropylene into Performance-Enhancing Warm-Mix Modifier for Asphalt Pavement Incorporating Recycled Concrete Aggregates" is carefully written and is well within the scope of the journal Polymers. Furthermore, it captures the relevance of sustainability and presents novel results relevant to the field. In my humble opinion, the manuscript is of good to quality to be considered for publication. My only suggestion is to improve the reporting on statistical information of the data, such as including number of testing parallels, Standard Deviation values, and checking the significant digits throughout the manuscript.

//Minor Revision

Reviewer 3 Report

Comments and Suggestions for Authors

The article is good. However, some minor concerns should be addressed before consider for publication

1.     The title should be revised by decreasing the numbers of words

2.     The grammatical and punctuation errors should be addressed to improve the overall readability and professionalism of the text. Ensure that the language is clear, concise, and easy to understand for the intended audience.

3.     Row 121 – 125 need to be explained in details

4.     The procedure and the variables (speed and temperature) mentioned in the section (Preparation of PP modified asphalt binders) and Figure 4 should be justified properly or at least supported by references.

5.     Results and discussion section is not sufficient. The authors can add more details and explain the reasons behind the results.

6.     Consider discussing the limitations of the study. This can help readers understand the scope and generalizability of the conclusions and recommendations.

7.     It is suggested to include a section on future research directions. This can provide insights into potential areas for further development in this field.

8.     The authors have to add some details about the possibility of industrial application for this article

Comments on the Quality of English Language

Moderate editing of English language required.

Round 2

Reviewer 1 Report

Comments and Suggestions for Authors

The authors addressed all my concerns, this manuscript is good for publication.

Reviewer 3 Report

Comments and Suggestions for Authors

No further comments